# EFFICIENT OCR FOR BUILDING A DIVERSE DIGITAL HISTORY

## ABSTRACT

**Abstract:** Many users consult digital archives daily, but the information they can access is unrepresentative of the diversity of documentary history. The sequence-to-sequence architecture typically used for optical character recognition (OCR) – which jointly learns a vision and language model - is poorly extensible to low-resource document collections, as learning a language-vision model requires extensive labeled sequences and compute. This study models OCR as a character level image retrieval problem, using a contrastively trained vision encoder. Because the model only learns characters' visual features, it is more sample efficient and extensible than existing architectures, enabling accurate OCR in settings where existing solutions fail. Crucially, it opens new avenues for community engagement in making digital history more representative of documentary history.

Digital texts are central to the study, dissemination, and preservation of human knowledge. Tens of thousands of users consult digital archives daily in Europe alone (Chiron et al., 2017), yet billions of documents remain trapped in hard copy in libraries and archives around the world. These documents contain extremely diverse character sets, languages, fonts or handwriting, printing technologies, and artifacts from scanning and aging. Converting them into machine-readable data that can power indexing and search, computational textual analyses, and statistical analyses - and be more easily consumed by the public - requires highly extensible, accurate, efficient tools for optical character recognition (OCR).

Current OCR technology - developed largely for small-scale commercial applications in high resource languages - falls short of these requirements. OCR is typically modeled as a sequence-to-sequence (seq2seq) problem, with learned embeddings from a neural vision model taken as inputs to a learned neural language model. The seq2seq architecture is challenging to extend and customize to novel, lower resource settings (Hedderich et al., 2021), because training a vision-language model requires a vast collection of labeled image-text pairs and significant compute. This study shows that on printed Japanese documents from the 1950s, the best performing existing OCR mis-predicts over half of characters. Poor performance is widespread, spurring a large post-OCR error correction literature (Lyu et al., 2021; Nguyen et al., 2021; van Strien. et al., 2020) and skewing digital history towards limited settings that are not representative of the diversity of documentary history.

This study develops a novel, open source OCR architecture, EffOCR (**Eff**icient**OCR**), designed for researchers and archives seeking a sample-efficient, customizable, scalable OCR solution for diverse documents. EffOCR combines the simplicity of early OCR systems, such as Tauschek's 1920s reading machine, with deep learning, bringing OCR back to its roots: the *optical* recognition of *characters*. Deep learning-based object detection methods are used to localize individual characters or words in the document image. Character (word) recognition is modeled as an image retrieval problem, using a vision encoder contrastively trained on character (word) crops.

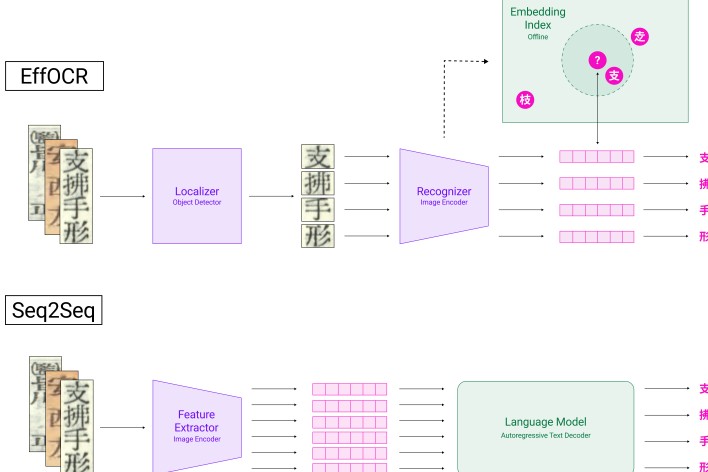

Figure 1: **EffOCR and Seq2Seq Model Architectures.** This figure represents the EffOCR architecture, as compared to a typical sequence-to-sequence OCR architecture.

EffOCR performs very accurately, even when using lightweight models designed for mobile phones that are cheap to train and deploy. Using documents that are fundamental to studying Japan's remarkable 20th century economic growth, the study shows EffOCR can provide a sample efficient, highly accurate OCR architecture for contexts where all current solutions fail. EffOCR's blend of accuracy and efficient runtime also makes it attractive for digitizing massive-scale collections in high resource languages, which the study illustrates with Library of Congress's collection of historical U.S. newspapers (Library of Congress, 2022). EffOCR has been used to cheaply and accurately digitize the over 20 million page scans in this collection (Dell et al., forthcoming).

In principle, contextual understanding could be extremely valuable to OCR, but in practice state-of-the-art transformer seq2seq models are extremely costly to train, expensive to deploy, and do not exist for lower resource languages, with advances concentrated in a handful of languages. This study shows that taking a step back from seq2seq models unlocks massive gains in sample efficiency. Researchers, with a modest number of annotations and modest compute, can tune their own OCR for settings where all existing solutions fail, using our user-friendly EffOCR open-source package. New characters specific to a setting can also be added at inference time - since they don't need to be seen in sequence during training - important for contexts such as archaeology where new characters are regularly discovered. These features facilitate making digital history more representative of documentary history.

## METHODS

Modern OCR overwhelmingly uses deep neural networks - either a convolutional neural network (CNN) or vision transformer (ViT) - to encode images. The representations created by passing an input image through a neural encoder are then decoded to the associated text.

Figure 1 underscores two fundamental differences between EffOCR and seq2seq. First, sequence-to-sequence architectures typically require line level inputs, and individual characters or words are not localized; rather, images or their representations are divided into fixed size patches. In contrast, EffOCR localizes characters and words using modern object detection methods (Cai & Vasconcelos, 2018; Jocher,

2020). Second, seq2seq sequentially decodes the learned image representations into text using a learned language model that takes the image representations as inputs. In contrast, EffOCR recognizes text by using contrastive training (Khosla et al., 2020) to learn a meaningful metric space for character or word-level OCR. The vision encoder projects crops of the same character (word) - regardless of style - nearby, whereas crops of different characters (words) are projected further apart. Character (word) embeddings are decoded to text in parallel by retrieving their nearest neighbor in an offline index of exemplar character (word) embeddings, created by rendering character (word) images with a digital font. Distances are computed using cosine similarity with a Facebook Artificial Intelligence Similarly Search (FAISS) backend (Johnson et al., 2019). The vision embeddings alone are sufficient to infer text since they represent characters - not text lines like in seq2seq - and hence decoding them does not require a language model or learned parameters.

This study develops both character and word level OCR models, with the former being more suitable for character-based languages and the latter more suitable for alphabet-based languages. When modeling OCR as a word level problem, EffOCR defaults to character level recognition if the distance between a word crop embedding and the nearest embedding in the offline dictionary of word embeddings is below a threshold cosine similarity. This is important, as hyphenated words at the end of lines, acronyms, proper nouns, and antiquated terms often make it infeasible to construct a comprehensive word dictionary.

EffOCR is trained on digital font renders, along with a modest number of labeled crops from the target datasets. The EffOCR recognizer is trained using the Supervised Contrastive ("SupCon") loss function (Khosla et al., 2020), a generalization of the InfoNCE loss (Oord et al., 2018) that allows for multiple positive and negative pairs for a given anchor. We use the "outside" SupCon loss formulation:

$$\mathcal{L}_{\text{out}}^{\text{sup}} = \sum_{i \in I} \mathcal{L}_{\text{out},i}^{\text{sup}} = \sum_{i \in I} \frac{-1}{|P(i)|} \sum_{p \in P(i)} \log \frac{\exp(\boldsymbol{z}_i \cdot \boldsymbol{z}_p / \tau)}{\sum_{a \in A(i)} \exp(\boldsymbol{z}_i \cdot \boldsymbol{z}_a / \tau)}$$

as implemented in PyTorch Metric Learning (Musgrave et al., 2020), where $\tau$ is a temperature parameter, $i$ indexes a sample in a "multiviewed" batch (in this case multiple fonts/augmentations of characters with the same identity), $P(i)$ is the set of indices of all positives in the multiviewed batch that are distinct from $i$, $A(i)$ is the set of all indices excluding $i$, and $z$ is an embedding of a sample in the batch.

To create training batches for the recognizer, EffOCR uses a custom $m$ per class sampling algorithm without replacement. This metric learning batch sampling algorithm also implements batching and training with hard negatives, where the negative samples in a batch are selected to be semantically close to one another, and thus contrasts made between anchors and hard negatives may be especially informative.

Different vision encoders can be used interchangeably for the EffOCR character localizer - which locates the character/word crops - and recognizer - which learns a metric space for these crops. Three models are considered for character level EffOCR: a vision transformer model (EffOCR-T Base) with XCiT (Small) (Ali et al., 2021) for both the localizer and recognizer, a convolutional base model (EffOCR-C Base) with ConvNeXt (Tiny) (Liu et al., 2022) for both the localizer and recognizer, and a convolutional small model (EffOCR-C Small), which uses lightweight architectures designed for mobile phones - YOLOv5 (Small) (Jocher, 2020) for the localizer and MobileNetV3 (Small) for the recognizer. For word level OCR, we develop EffOCR-Word Small, which uses the same lightweight architectures as EffOCR-C Small. EffOCR-Word Small defaults to EffOCR-C Small when the cosine similarity between a word crop embedding and the nearest embedding in the offline word embedding dictionary is below 0.82, a hyperparameter tuned on the validation set. The base models use a two-stage object detector for character localization, specifically a Cascade R-CNN (Cai & Vasconcelos, 2019), whereas the small models use one-stage object detection for faster speed (Jocher, 2020). The supplementary materials describe the EffOCR architecture and training recipes with no detail spared and evaluate models using alternative vision transformer encoders.

EffOCR's architecture draws inspiration from metric learning methods for efficient image retrieval (El-Nouby et al., 2021), joining a recent literature on self-supervision through simple data augmentation for image encoders (Grill et al., 2020; Chen et al., 2021; Chen & He, 2021). The closest frameworks to EffOCR

in their overall design are the original OCR conceptualizations, such as Tauschek's 1920s reading machine, which used human engineered features to recognize localized characters. More recently, CharNet (Xing et al., 2019), developed for scene text (not documents), uses separate convolutional networks for dense classification and regression at a single scale, outputting a character class and bounding box at every spatial location, and then aggregates this information with confidence scores to make final predictions. EffOCR in contrast deploys widely used, highly optimized object detection methods to localize characters and then feeds character crops to a contrastively trained recognizer.[1]

## TRAINING AND EVALUATION DATASETS

Evaluating EffOCR requires benchmark datasets that are representative of the diversity of documentary history. Traditional OCR benchmarks focus on commercial applications like receipts (Huang et al., 2019) - and SOTA OCR systems evaluate on these data - which are not relevant to digital history.

Instead, the study draws on the literature on historical image datasets (Nikolaidou et al., 2022). First, it uses documents from historical Japan that can elucidate fundamental questions that have been understudied due to a lack of digital data, such as the drivers of Japan's rapid transformation from a poor agrarian economy to a wealthy industrialized nation. Horizontally and vertically written tabular data - providing rich information on Japanese firms and their personnel - are drawn from two 1950s publications (Jinji Koshinjo, 1954; Teikoku Koshinjo, 1957). A 1930s prose publication providing detailed biographies of tens of thousands of individuals (Jinji Koshinjo, 1939) is also examined. These texts could use over 13,000 *kanji* characters.

The second context is Library of Congress's Chronicling America (LoCCA) collection, which contains over 19 million historical newspaper page scans. This collection is highly diverse, as shown in Figure 2.

| | |
|---|---|
| WASHINGTON, April 1—Ambas- | WASHINGTON, April 1 Ambas- |
| FORT WORTH JITNEYS QUIT | FORT WORTH JITNEYS OUIT |
| General Plan  5- 4-31 | General Plan  5-4-31 |
| State of Tennessee, | State of Tennessee |
| A non-Federal project to furnish free home assistance | A non-Federal project to furnish free home assistance |
| SEED DISTRIBUTION | SEED DISTRIBUTION |
| Iron. Steel and Tin Workers | Iron, Steel and Tin Workers |
| ADVERSE REPORTS ON DEMENT'S NOMINATION. | ADVERSE REPORTS ON DEMENTS NOMINATION |
| IMPROVEMENT IS SHOWN | IMPROVEMENT IS SHOWN |

Figure 2: **Diversity in the Chronicling America Dataset.** This figure shows examples sampled from the Chronicling America (LoCCA) dataset, along with EffOCR predicted transcriptions.

Library of Congress provides an OCR, but the quality is low (Smith et al., 2015). There is a large literature studying historical newspapers at scale, which overwhelmingly uses keyword search and does not unlock the power of large language models due to poor quality digitization (Hanlon & Beach, 2022). LoCCA elucidates how EffOCR: 1) performs in the highest resource setting, English; 2) extensibility across Latin

---

[1]Others have also used contrastive learning for OCR, in particular (Aberdam et al., 2021) use a self-supervised, sequence-to-sequence contrastive learning approach.

and *kanji* characters, which differ significantly in their aspect ratios and complexity; 3) extensibility to the many Unicode renderable languages that use the Latin script.

Layout datasets exist for Chronicling America and some of the Japanese publications (Shen et al., 2020; Lee et al., 2020). Adding word/character bounding boxes and transcription annotations builds upon the existing work of the historical image dataset literature (Nikolaidou et al., 2022). Because seq2seq requires lines as inputs, to build the Japanese and Chronicling America datasets we draw lines at random from the Japanese volumes and from 10 randomly selected newspapers in LoCCA. Lines correspond to cells in tables and single lines within columns/rows in prose. The baseline training sets range from 291 lines for Chronicling America to 1309 cells for horizontal Japanese, highly feasible for researchers to label in an afternoon, and also includes val and test splits.

For the newspapers, we also provide an additional evaluation-only dataset that consists of a sample of 225 textlines, randomly drawn from all scans in the Chronicling America collection published on March 1st of years ending in "6," from 1856-1926. This sample is balanced across these decades, with 25 textlines sampled randomly from each of the days. A selection of textlines from this set is shown in Figure 2. The day-per-decade set is designed to be challenging, by weighting older, much harder to read scans from the mid-19th century equally despite their relative scarcity in the Chronicling America collection.

In addition to this gold quality training and evaluation data, we create silver quality training data for training EffOCR-Word (Small) by applying the EffOCR-C (Small) model to a random sample of newspapers. We limited the number of crops with model-generated labels to 20 - so each word can have 0-20 silver-quality crops depending upon its frequency of occurrence in our random sample. This limit is binding for common words, *e.g.,* "the". We also use the gold word crops from the 291 line training set, which cover only a small share of words that could appear. Using silver quality data leads to high performance, achieved essentially for free. The study's training datasets are publicly released.

Finally, we examine EffOCR on an existing Polytonic Greek benchmark (Gatos et al., 2015), selected because it contains both line-level and word transcriptions. Polytonic Greek uses five diacritics to notate older Greek texts. It is challenging because the diacritics have a similar appearance. The supplemental materials show example documents from all benchmarks.

## MEASUREMENT AND COMPARISONS

OCR accuracy is measured using the character error rate (CER), the Levenshtein distance between the OCR'ed string and the ground truth, normalized by the length of the ground truth. A CER of 0.5, for instance, translates to mispredicting approximately half of characters.

The most widely used OCR engines are commercial products that do not support fine-tuning and have proprietary architectures. The study compares EffOCR to Google Cloud Vision (GCV) and Baidu OCR (popular for Asian languages). We also consider four open source architectures: EasyOCR's convolutional recurrent neural network (CRNN) framework (Shi et al., 2016), TrOCR's sequence-to-sequence encoder-decoder transformer (base and small) (Li et al., 2021), Tesseract's bi-directional LSTM, and PaddleOCR's Single Vision Text Recognition (SVTR), which also abandons seq2seq, dividing text images into small (non-character) patches, using mixing blocks to perceive inter- and intra-character patterns, and recognizing text by linear prediction (Du et al., 2022). A large literature has examined a variety of custom-designed OCR systems. We focus on those that either (1) make similar architectural choices (SVTR), (2) are considered SOTA, regardless of architectural choices (TrOCR), or (3) are very popular (Tesseract and EasyOCR).

The pre-trained EasyOCR, PaddleOCR, and TrOCR models are fine-tuned on the same target data as EffOCR. Considerable resources have been devoted to pre-training these models. For example, TrOCR was pre-trained on 684 million English synthetic text lines. Hence, these comparisons elucidate performance

when these pre-trained models are further tuned on the target datasets. For a more apples-to-apples comparison, the study examines the accuracy of these architectures when trained from scratch (using a pre-trained checkpoint not trained for OCR, when supported by the architecture) on 8,000 synthetic text lines (like EffOCR) and the same target crops. EasyOCR and PaddleOCR do not support vertical Japanese, and TrOCR does not support any Japanese. Tesseract offered little support for fine-tuning until recently and hence most of its applications have been off-the-shelf, which is this study's focus.

## RESULTS

EffOCR provides a highly accurate OCR with minimal training data, in contexts where current solutions fail. For vertical Japanese tables, the best EffOCR CER is 0.7% (Table 1). The next best alternative, Baidu OCR, has a CER of 55.6%, making nearly 80 times more errors. The best EffOCR CER is modestly higher for the Japanese prose (2.7%); these scans are low resolution and some characters are illegible, to provide a context where OCR with language modeling could offer a clear advantage. Yet EffOCR makes 5 times fewer errors than the next best alternative (GCV), whose CER of 13.5% will not support applications that require high accuracy. For horizontal Japanese - a higher resource setting - the EffOCR CER is 0.6%, whereas the next-best-alternative (Paddle OCR fine-tuned on target crops) makes more than five times more errors. The different EffOCR models produce strikingly similar results, despite the significant differences in architecture (convolutional versus transformer) and model size (9.3M to 112.5M parameters).

| Model/Engine | Seq2Seq? | Transformer? | Pretraining | Parameters | Character Error Rate | | | | | | Lines/second | |
| | | | | | Horiz. Jap. | Vertical Jap. (tables) | Vertical Jap. (prose) | Chron. Amer. Eval | Chron. Amer. Day/Decade | Anci. Greek | Horiz. Jap. | Chron. Amer. |
|---|---|---|---|---|---|---|---|---|---|---|---|---|
| EffOCR-C (Base) | × | × | from scratch | 112.5 M | **0.006** | **0.007** | 0.030 | 0.023 | 0.062 | 0.049 | 0.79 | 0.49 |
| EffOCR-C (Small) | × | × | from scratch | 9.3 M | 0.010 | 0.009 | 0.036 | 0.028 | 0.080 | 0.052 | 19.46 | 13.40 |
| EffOCR-T (Base) | × | ✓ | from scratch | 101.8 M | 0.009 | 0.007 | **0.027** | 0.022 | 0.059 | **0.047** | 0.19 | 0.31 |
| EffOCR-Word (Small) | × | × | from scratch | 10.6 M | - | - | - | 0.015 | 0.043 | - | - | 21.36 |
| Google Cloud Vision OCR | ? | ? | off-the-shelf | ? | 0.173 | 0.695 | 0.135 | **0.005** | **0.019** | 0.065 | ? | ? |
| Baidu OCR | ? | ? | off-the-shelf | ? | 0.060 | 0.556 | 0.177 | - | - | - | ? | ? |
| Tesseract OCR (Best) | ✓ | × | off-the-shelf | 1.4 M | 1.021 | 0.996 | 0.744 | 0.106 | 0.170 | 0.251 | 4.90 | 4.47 |
| EasyOCR CRNN | ✓ | × | off-the-shelf | 3.8 M | 0.191 | - | - | 0.170 | 0.274 | - | 33.55 | 19.80 |
| | | | fine-tuned | | 0.082 | - | - | 0.036 | 0.157 | | | |
| | | | from scratch | | 0.132 | - | - | 0.131 | 0.204 | | | |
| PaddleOCR SVTR | × | × | off-the-shelf | 11 M | 0.085 | - | - | 0.304 | 0.314 | - | 13.34 | 13.56 |
| | | | fine-tuned | | 0.032 | - | - | 0.103 | 0.129 | | | |
| | | | from scratch | | 0.097 | - | - | 0.104 | 0.138 | | | |
| TrOCR (Base) | ✓ | ✓ | off-the-shelf | 334 M | - | - | - | 0.015 | 0.038 | - | - | 0.43 |
| | | | fine-tuned | | - | - | - | 0.013 | 0.027 | | | |
| | | | from scratch | | - | - | - | 0.809 | 0.831 | | | |
| TrOCR (Small) | ✓ | ✓ | off-the-shelf | 62 M | - | - | - | 0.039 | 0.121 | - | - | 0.97 |
| | | | fine-tuned | | - | - | - | 0.075 | 0.091 | | | |
| | | | from scratch | | - | - | - | 0.773 | 0.820 | | | |

Table 1: **Baseline Results and Comparisons.** This table reports the performance of different OCR architectures, *off-the-shelf* (without fine-tuning on target data), *fine-tuned* on the target publication training set from a pre-trained OCR checkpoint, and trained *from scratch* on synthetic text lines and the target publication training set. "?" indicates that the field is unknown due to the proprietary nature of the architecture.

The CER (uncased) for the LoCCA newspapers is 1.5%. GCV has the best performance (0.5%), followed by fine-tuned TrOCR (Base) (1.3% CER). The advantage of EffOCR on English - the quintessential high resource setting - is its open-source codebase and fast runtime. GCV makes significant layout errors when fed full newspaper page scans, which have complex layouts (Shen et al., 2021), and hence the performance in

Table 1 cannot be replicated when it is fed scans. GCV charges per image, and the supplementary materials estimate a cost at current prices of $23 million USD to digitize LoCCA at the line image level, versus $60K for EffOCR-Word (Small).

Table 1 examines CPU runtime for open source architectures, measured by lines processed per second on identical dedicated hardware (GPUs are prohibitively costly for mass digitization). EffOCR-Word (Small) is 50 times faster than TrOCR (Base), which is likely to be cost prohibitive for larger scale applications. EffOCR supports inference parallelization across characters - promoting faster inference - whereas seq2seq requires autoregressive decoding. On English, the most plausible scalable alternative is fine-tuned EasyOCR. With a third of the parameters of EffOCR-Word (Small), inference is faster, but the CER is around 29% higher. For horizontal Japanese, EffOCR-C (Small) is three times more accurate and faster than PaddleOCR SVTR (fine-tuned), the next best alternative.

Figure 3: **Error Analysis.** Representative examples of EffOCR errors, showing the target crop, the EffOCR localized crop, and the five nearest characters in the embedding index, with the correct character highlighted in green.

Figure 3 provides representative examples of errors, showing the target crop, the localized crop, and its five nearest neighbors, with the correct prediction highlighted in green. Errors tend to occur when the character is illegible or homoglyphic to another character (*e.g.* O and 0). For example, a 0 in one font can occasionally be indistinguishable from an O in another, an error that would be straightforward to correct in post-processing. The supplementary materials report results from additional encoders, and examine how different ingredients of EffOCR contribute to its performance.

EffOCR outperforms all other architectures that support Polytonic Greek, including Google Cloud Vision. This illustrates the versatility of the architecture.

EffOCR's parsimonious architecture allows it to learn efficiently. To quantify this, we train different OCR models from scratch using varying amounts of annotated data. All architectures are pre-trained from scratch on 8,000 synthetic text lines, starting from pre-trained checkpoints not customized for OCR when supported by the framework. They are then fine-tuned on the study's benchmark datasets, with varying train splits: 70%, 50%, 20%, 5%, and 0% (using only synthetic data). These exercises are performed for Chronicling America and horizontal Japanese, as vertical Japanese is not supported by the comparison architectures.

Figure 4 plots the percentage of the benchmark dataset used in training on the x-axis and the CER on the y-axis. On just 99 labeled table cells for Japanese and 21 labeled rows for LoCCA (the 5% train split), EffOCR's CER is around 4%, showing viable few shot performance. The other architectures remain unusable. EffOCR performs nearly as well using 20% of the training data as using 70%, where it continues to outperform all other alternatives.

The focus of this study is on developing an architecture that is sample efficient to customize to highly diverse settings where existing solutions do not provide the desired accuracy, or do not do so within budget

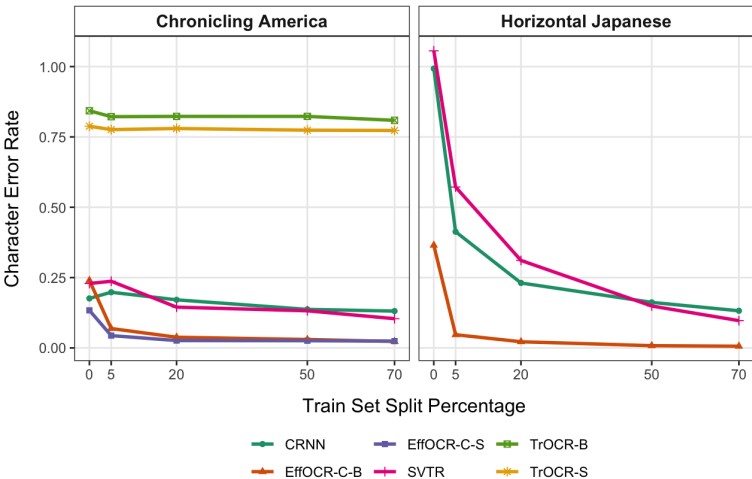

Figure 4: **Sample Efficiency.** This figure plots the percentage of the benchmark dataset used in training against the character error rate, for different OCR model architectures.

constraints. We have not aimed to create a broadly applicable off-the-shelf solution, but nevertheless evaluate how EffOCR does fully out-of-distribution by comparing EffOCR-Word (Small) to other solutions on a highly diverse dataset sampled randomly from 64 randomly selected document record groups in the U.S. National Archives. This is a challenging dataset, with examples shown in the supplementary materials. The supplementary materials show that EffOCR-Word (Small) performs similarly to other open-source OCR engines (CER = 12.6), having only been exposed to highly out-of-distribution newspapers during training. The sample efficiency of EffOCR suggests it could be trained to perform even better off-the-shelf on diverse archival documents by labeling a small number of samples across a wide range of common historical document types. We plan to crowd source this effort in the future.

## LIMITATIONS

This study does not focus on handwriting due to space constraints, but the approach would be analogous. Synthetic handwriting generators, *e.g.* (Bhunia et al., 2021), could provide extensive data for pre-training, analogous to this study's use of digital fonts. There are some settings where EffOCR's framework is not suitable. If large portions of a document are illegible, context is necessary. Moreover, the heavy use of ligatures and/or slanting in some character sets and handwriting could lead to more challenging character localization. This problem is addressed with the word-level EffOCR model.

## DISCUSSION

Indexing, analyzing, disseminating, and preserving diverse documentary history requires community engagement of stakeholders with the requisite fine-grained knowledge of the relevant settings. EffOCR facilitates this engagement because it is highly extensible to low-resource settings, sample-efficient to customize, and simple and cheap to train and deploy. In contrast, seq2seq is more aligned with the commercial objective of designing a product that is difficult for competitors to imitate. For example, EffOCR can be trained in the

cloud with free student compute credits, whereas TrOCR required training on a multi-million dollar cluster with 32 32GB V100 cards. Lower resource languages may lack the pre-trained large language models required to initialize a transformer seq2seq model like TrOCR, and compute resources and data for training are also likely unavailable. EffOCR encourages community engagement by combining the parsimonious conceptualization of OCR from nearly a century ago with deep learning, integrating the follow features:

**Character/word level**: EffOCR creates semantically rich visual embeddings of individual characters (words), a parsimonious problem. Annotators can select which of the most probable character (word) predictions from the pre-trained recognizer are correct, potentially using a simple mobile interface, or line level labels can be mapped to the character (word) level once a localizer has been developed.

**Language Extensibility**: Language modeling advances have concentrated around less than two dozen modern languages, out of many thousands (Joshi et al., 2020). Omitting the language model makes EffOCR extensible and easy-to-train. To extend EffOCR to a new language, all one needs are renders for the appropriate character set. Additionally, characters do not need to be seen in sequence during training, so new characters can be added at inference time, valuable for archaeological contexts where new characters are regularly discovered. Omitting the language model makes it easy to mix scripts, necessary for some languages. The recognizer can also be exposed to characters in training using any desired sequencing. This is not true of multilingual seq2seq training, which leads to many OCR errors with endangered languages (Rijhwani et al., 2020). EffOCR can convert the Japanese publications examined in this study into a knowledge graph, revealing rich economic insights about Japanese economic development that were previously unknown due to the failure of all existing OCR solutions. The supplementary materials provide more details.

**Decoupling localization and recognition:** Theoretically, localization and recognition (akin to classification) may rely on different features of the image, suggesting modularity (Song et al., 2020). Practically, decoupling allows localization and recognition to use different training sets, economizing on annotation costs since these tasks can require very different numbers of labels depending on the script. It also encourages community innovation and future-proofness, because it simplifies training recipes and makes it straightforward to swap in new localizers or recognizers - including zero-shot models such as Kirillov et al. (2023) - as the literature advances.

**Scalable:** The small EffOCR models achieve fast CPU inference that can scale cheaply to hundreds of millions of documents. EffOCR-Word (Small) has been used to digitize the approximately 20 million scans in the Chronicling America collection on a $60,000 budget for inference, whereas TrOCR would have cost nearly 50 times as much and GCV would have been orders of magnitude more expensive.

**Open-Source:** We have also released EffOCR as a user-friendly, open-source python package. The EffOCR package makes it straightforward to use existing EffOCR models off-the-shelf with just a few lines of code. It also includes functionality to train custom models and guides users with detailed tutorials. By engaging the community - including those who lack extensive experience with deep learning frameworks - we hope to make digital history more representative of the diversity of human history.

## REPRODUCIBILITY

We release all code and training data used to create EffOCR. Scripts in the public repository exactly reproduce the figures cited above. All other material needed to reproduce these results is detailed in the supplemental materials, including training hyperparmeters. The models in this paper can also be deployed through the EffOCR python package.

ETHICS

EffOCR presents no major ethical concerns. Its methods are entirely open source, and its training data are entirely in the public domain. Its core functionality, accurately transcribing text in low-resource settings, is ethically sound. Some applications of EffOCR could raise ethical flags. We discourage users from applying EffOCR to copyrighted documents unless the application is protected by fair use. While EffOCR is a potentially useful tool for studying bias and/or harmful content, harmful content transcribed by EffOCR should not be shared without proper context.

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

# Supplementary materials

**Anonymous authors**

## Materials and Methods

### Encoders

Different encoders can be used interchangeably for EffOCR's character localization module (hereafter, "localizer") and character recognizing module (hereafter "recognizer"). We use the following:

- **EffOCR-C (Base)**: ConvNeXt (Tiny) (Liu et al., 2022) for both the localizer and recognizer. Both models are initialized from the officially released checkpoint with specifications:
  `{size: "tiny"}`
- **EffOCR-T (Base)**: XCiT (Small) (Ali et al., 2021) for both the localizer and recognizer. Both models are initialized from the officially released checkpoint with specifications:
  `{size: "small", depth: 12, patch_size: 8, resoultion: 224}`
- **EffOCR-C (Small) and EffOCR-Word (Small)**: YOLOv5 (Small) (Jocher, 2020) for the localizer and MobileNetV3 (Small) for the recognizer. YOLOv5 is initialized from the officially released `YOLOv5s` checkpoint, and MobileNetV3 is initially from the PyTorch Image Models ("timm") (Wightman, 2019) produced checkpoint with specifications:
  `{size: "small", channel_multiplier: 0.50}`

For ablations, we also examine:

- Swin (Tiny) (Liu et al., 2021) for both the localizer and recognizer. Both models are initialized from the officially released checkpoint with specifications:
  `{size: "tiny", patch_size: 4, window: 7, resolution: 224}`
- ViTDet (Base) (Li et al., 2022) for the localizer and a vanilla vision transformer, ViT (Base), for the recognizer. Both models are initialized from the officially released checkpoint with specifications:
  `{size: "base", patch_size: 16, resolution: 224}`

These architectures were selected for the following reasons:

- **EffOCR-C (Base)**: ConvNeXt is a new state-of-the-art CNN backbone, in contrast to the other three vision transformer encoders.
- **EffOCR-T (Base)**: XCiT was chosen because of its comparative advantage in modeling fine-grained features via the ability to accommodate smaller patch sizes through a linear complexity attention mechanism, which may be especially suitable for character images with small spatial extents (as measured in pixels).
- **EffOCR-C (Small) and EffOCR-Word (Small)**: MobileNetV3 (Small) and YOLOv5 (Small) were collectively chosen to produce a speed optimized EffOCR, as both architectures are popular, easily customizable, and speed-optimized by design.

- The Swin transformer was selected because of its state-of-the-art performance on object detection tasks.

- The original ViT embeddings perform well for image retrieval, and have become a new baseline for image retrieval (El-Nouby et al., 2021).

The inference speed advantages offered by a smaller transformer encoder, such as MobileViT, are much more modest than that offered by MobileNetV3, and hence an EffOCR-T (small) model is not developed, although it would be straightforward to do so should users desire it. In tests, a MobileViTv2 (small) Recognizer model was approximately 6.5 times slower than a comparable MobileNetv3 Recognizer.

As the deep learning literature advances and new models are developed, EffOCR's modular framework and simple training recipes make it straightforward to swap in new encoders, granting the model a degree of future-proofness.

These models are all trained on a single A6000 GPU card, with hyperparameters selected using the 15% validation split, save for the models with XCiT (Small) or ViT (Base) encoders, which were trained on two A6000 GPU cards.

CHARACTER LOCALIZATION

All models use an MMDetection (Chen et al., 2019) backend for localization, except for the ViTDet ablation, which uses Detectron2 (Wu et al., 2019) and YOLOv5 (Small) (Jocher, 2020) for EffOCR-C (Small), which uses its own custom training scripts. Only one EffOCR configuration, EffOCR-C (Small), has a localizer that uses a one-stage object detection framework: YOLOv5 (Small) (Jocher, 2020). All others use a two-stage object detector, specifically a Cascade R-CNN (Cai & Vasconcelos, 2019). One stage object detection is faster, and hence makes sense for the small model, where a central objective is fast inference speed.

The localizers built with ConvNeXt (EffOCR-C Base), XCiT (EffOCR-T Base), and Swin (ablation) are trained on 8,000 textlines of synthetic data for 40 epochs at a constant learning rate of $1e-4$ and fine-tuned on benchmark data for 100 epochs at a $2.5e-5$ constant learning rate, all with anchor generator scales $[2, 8, 32]$. ViTDet is trained on 8,000 textlines of synthetic data for 40 epochs with a constant learning rate of $1e-4$, and then fine-tuned for 100 epochs on benchmark data with a $1e-5$ constant learning rate. The YOLO localizer is trained on 8,000 textlines of synthetic data for 30 epochs at a constant learning rate of $1e-2$ and fine-tuned on benchmark data for 30 additional epochs, still at a constant $1e-2$ learning rate.

The synthetic data used for pre-training the localizers and comparison models was created using a custom synthetic data generator.

This generator was used to create six synthetic dataset variants, each consisting of 10,000 synthetic lines with an 80%-10%-10% train-test-validation split. The six dataset variants are: horizontal English with character sequences generated at random, horizontal Japanese with character sequences generated at random, vertical Japanese with character sequences generated at random, horizontal English with text sequences generated from Wikipedia, horizontal Japanese with text sequences generated from (Japanese) Wikipedia, and vertical Japanese with text sequences generated from (Japanese) Wikipedia. Localizers for detecting Greek text were pretrained on synthetic English data due to broad similarities between lines. Text sequence based synthetic datasets were used to pre-train seq2seq models that rely on language context, e.g., TrOCR and CRNN; character sequence based synthetic datasets were used to pre-train non-seq2seq models, e.g., EffOCR and SVTR.

CHARACTER RECOGNITION

The EffOCR recognizer is trained using the Supervised Contrastive ("SupCon") loss function (Khosla et al., 2020), a generalization of the InfoNCE loss (Oord et al., 2018) that allows for multiple positive and negative pairs for a given anchor. In particular, we work with the "outside" SupCon loss formulation

$$\mathcal{L}_{\text{out}}^{\text{sup}} = \sum_{i \in I} \mathcal{L}_{\text{out},i}^{\text{sup}} = \sum_{i \in I} \frac{-1}{|P(i)|} \sum_{p \in P(i)} \log \frac{\exp(z_i \cdot z_p / \tau)}{\sum_{a \in A(i)} \exp(z_i \cdot z_a / \tau)}$$

as implemented in PyTorch Metric Learning (Musgrave et al., 2020), where $\tau$ is a temperature parameter, $i$ indexes a sample in a "multiviewed" batch (in this case multiple fonts/augmentations of characters with the same identity), $P(i)$ is the set of indices of all positives in the multiviewed batch that are distinct from $i$, $A(i)$ is the set of all indices excluding $i$, and $z$ is an embedding of a sample in the batch (Khosla et al., 2020).

To create training batches for the recognizer, EffOCR uses a custom $m$ per class sampling algorithm *without replacement* adapted from the PyTorch Metric Learning repository (Musgrave et al., 2020). This metric learning batch sampling algorithm also implements batching and training with hard negatives, where the negative samples in a batch are selected to be semantically close to one another, and thus contrasts made between anchors and hard negatives may be especially informative for the model to update on. Indeed, one of the main advantages of contrastive training is that it allows the learning process to exploit hard negative mining.

More specifically, the custom batch sampling algorithm samples $m$ character variants for each class (character) - drawn from both target documents and augmented digital fonts. We choose $m = 4$ and the batch size is 128, meaning 4 styles/representations of each of 32 different characters appear in each batch. The model learns to map character crops of the same identity to similar dense vectors in a semantically rich, high-dimensional vector space, and vice versa. There is no natural definition of an epoch in the context of batch-based sampling for contrastive learning with data augmentation in the way EffOCR formulates this procedure. For EffOCR recognizer training, an epoch is defined as some number $P$ passes through all unique characters $N$ in the character set under consideration, i.e., $N = 13,738$ for Japanese, $N = 91$ for English, and $N = 186$ for Polytonic Greek. Empirically, a good setting for Japanese is $P = 1$, so the total number of classes in an epoch is 13,738, for English $P = 10$, so the total number of classes in an epoch is 910, and for Greek $P = 4$, so the total number of classes in an epoch is 744. Sampling for each class occurs without replacement, for better coverage of character variants. Because of this, the number of passes $P$ matters, as it determines the number of character variants used for contrastive training in each epoch.

Every character crop that appears in the training set is embedded using a model first trained without hard negative mining/sampling, and for each we find its 8 nearest neighbors. The EffOCR recognizer is then trained again from scratch, with batches being sampled with an $m$ per class sampler (without replacement) that is further modified to randomly intersperse hard negative sets (8 nearest neighbor characters, $m = 4$ variants of each) throughout batches.

EffOCR is trained on digital font renders from readily available fonts (13 for Japanese, 14 for English, and 8 for Greek), along with a modest number of labeled crops from the target datasets.[1] The digital fonts

---

[1] Fonts for Japanese included: Dela Gothic One Regular; Hachi Maru Pop Regular; Hina Mincho Regular; Komorebi Gothic; Kosugi Regular; New Tegomin Regular; Noto Serif CJK JP Regular; Reggae One Regular; Shippori Mincho B1 Regular; Stick Regular; taisyokatujippoi7T5; Tanugo Regular; and Yomogi Regular. Fonts for English included: Anton Regular; Cutive Mono Regular; EB Garamond Regular; Fredoka Regular; IM Fell DW Pica Regular; NewYorker-jLv; Noto Serif Regular; Oldnewspapertypes-449D; Orbitron Regular; Special Elite Regular; Ultra Regular; VT323 Regular; ZaiConsulPolishTypewriter-MVAxw; and ZaiCourierPolski1941-Yza4q. Fonts for Polytonic Greek included EB Garamond Regular; Noto Serif Regular; SBL Greek; Gentium Book Plus Regular; Gentium Plus Regular; Gentium Plus Italic; Orbiton Regular; Ultra Regular; and

are augmented by randomly applying affine transformations (translation and scaling); background coloring, color jittering, color inversion, and grayscaling; and Gaussian blurring. The model trains on digital fonts and labeled crops *together*, since the objective is to learn general purpose embeddings that would map target crops nearby to digital renders. All recognizer models except MobileNetV3 use an AdamW optimizer with weight decay of $5e-4$, a SupCon loss with temperature of 0.1, a learning rate of $2e-5$, and a batch size of 128. MobileNetv3 uses the same parameters except a learning rate of $2e-3$. The Japanese datasets are trained for 60 epochs and the English and Greek datasets for 30.

After recognizer training is completed, the recognizer is used as an encoder to create an offline index of exemplar character embeddings to be searched at inference time for the purposes of character recognition. Specifically, the exemplar character embedding index is created by embedding image renders for all the unicode characters supported by the Google Noto Serif font series, i.e., Noto Serif CJK JP Regular for models trained for Japanese OCR and Noto Serif Regular for models trained for English and Greek OCR. The Google Noto series is chosen as an exemplar font due to both its extremely wide coverage of glyphs and the simplicity of its style, though, by virtue of EffOCR's training, other fonts could be used as well. At inference time, FAISS (Johnson et al., 2019) is used to perform an *inner product* similarity search that compares character embeddings in the sample being inferenced to exemplar character embeddings in this offline index; identities are assigned to inferenced characters using the identity of that character's nearest neighbor in the offline exemplar index, i.e., k-NN classification with $k = 1$.

For case sensitive applications, EffOCR character recognition for English text can also be lightly post-processed to help better differentiate uppercase and lowercase letters from one another: one can force a character to be uppercased or lowercased through simple rules based statistics about the dimensions of bounding boxes (in the sample undergoing inference). This procedure is irrelevant for results reported in this text, however, for which CER is measured uncased.

Greek text is also case-sensitive and CER from Greek data is also presented uncased, although upper- and lowercase Greek characters bear less resemblance than in English. Two additional rules were also applied when evaluating Greek text: apostrophes (') and accents (') were considered equivalent, and the *stigma* ligature was considered equivalent to the *terminal sigma* (ς) character.

Checkpoints/weights for all recognizers are supported by implementations from timm (Wightman, 2019).

### WORD RECOGNITION

We train word recognition as a nearest neighbor image retrieval problem. The training dataset for the model consists of digital renders of words created using 43 fonts, silver quality data from the target dataset created by applying the EffOCR-C (Small) model to a random sample of days, and a small number of randomly selected hand labeled word crops. We limited the number of crops with model-generated labels to 20 - so each word can have 0-20 silver-quality crops depending upon its frequency of occurrence in our random sample. This limit is binding for common words, *e.g.,* "the".

The recognizer is trained using the Supervised Contrastive ("SupCon") loss function (Khosla et al., 2020), as above. To create training batches for the recognizer, we use a custom $m$ per class sampling algorithm without replacement, adapted from the PyTorch Metric Learning repository (Musgrave et al., 2020). The $m$ word variants for each class (word) are drawn from both target documents and augmented digital fonts. We select $m = 4$ and the batch size is 1024, meaning 4 styles of each of 256 different words appear in each batch. For training without hard negatives, we define an epoch as letting the model see each word (case-sensitive) exactly $m = 4$ times. Sampling for each class occurs without replacement until all variants are exhausted.

In order to converge faster with limited compute, we also implement offline-hard negative mining, batching similar negatives and their corresponding positive anchors together - thus making the contrasts between the

positive and negative pairs within a batch especially informative. To create hard negative sets, we render each word using a reference font (Noto-Serif Regular) and embed it to create a reference index. We find $k = 8$ nearest neighbors for each word using this index and the model trained without hard negatives, which yields sets of 8 words that have a similar appearance when rendered with the reference font. We use only the reference font to create these sets because using crops corresponding to all 43 fonts for each word is computationally costly and creates more hard negative sets than we can use in training. We also use each word crop from the target dataset (both silver quality annotations generated with model predictions and gold quality human-annotated predictions) to create hard negative sets. Hence, the total number of hard-negative sets equals the number of words in our dictionary (generated with the reference font) plus the number of word crops from the newspaper data in the training set.

Each hard negative set contains 8 words, with $m = 4$ views per word, which means we can fit 32 randomly sampled hard negative sets within each batch. An epoch is defined as seeing each hard negative set once. Since the number of synthetic views of an image is much larger than the number of target newspaper crops, whenever newspaper crops are available we force the $m$ views of a word to contain an equal number of synthetic and target crops.

We use a MobileNetV3 (Small) encoder pre-trained on ImageNet1k sourced from the timm (Wightman, 2019) library, more specifically, the model *mobilenetv3_small_050*. We use 0.1 as the temperature for Sup-Con loss and AdamW as the optimizer with Pytorch defaults for all parameters other than weight decay (5e-4) and learning rate. We used Cosine Annealing with Warm Restarts as the learning rate scheduler with a maximum learning rate of $2e - 3$, a minimum learning rate of 0, time to first restart ($T_0$) as the number of batches in an epoch, and restart factor, $T_{mult}$ of 2 using the implementation provided in Pytorch.

While fonts and newspaper crops for each word act as an augmentation on the skeleton of the word, we also add more image-level transformations to improve generalization. These include Affine transformation (only slight translation and scaling allowed), Random Color Jitter, Random Autocontrast, Random Gaussian Blurring, Random Grayscale, Random Solarize, Random Sharpness, Random Invert, Random Equalize, Random Posterize and Randomly erasing a small number of pixels of the image. Additionally, we pad the word to make the image square while preserving the aspect ratio of the word render. We do not use common augmentations like Random Cropping or Center Cropping, to avoid destroying too much information.

The model trained without hard negatives was trained for 50 epochs and with hard negatives, it was trained for 40 epochs. For selecting the best checkpoint, we use 1-CER (OCR Character Error Rate) as the validation metric on the validation set. We chose the model that performed best in terms of CER when detecting only words on the validation set. This means that if a word is outside of our dictionary, it is forcefully matched to the nearest neighbor in the dictionary. The best model achieved a CER of 4.9% with word-only recognition.

At inference time, words are recognized by retrieving their nearest neighbor from the offline embedding index created with the reference font, using a Facebook Artificial Intelligence Similarity Search backend (Johnson et al., 2019). The code to train the model and generate training data, as well as the model checkpoints, are made publicly available.

### COMPARISONS

To examine sample efficiency, we train alternative architectures from scratch, on the same number of synthetic text lines used to train EffOCR. Specifically, the comparison architectures are, as applicable, initialized with "default" pre-trained checkpoints that have not yet been exposed to an OCR task, e.g., masked language model pre-trained weights for text transformers or ImageNet pre-trained weights for CNNs and vision transformers. These comparison architectures are then trained on 8,000 synthetic text lines per the applicable synthetic dataset variant (see: Methods - Synthetic Data) as a form of standardized OCR-task-specific pretraining. They are then fine-tuned on the same benchmark datasets used to assess EffOCR, but with varying

train-test-validation splits: 70%-15%-15%, 50%-25%-25%, 20%-40%-40%, 5%-47.5%-47.5%, and 0%-50%-50% (i.e., zero-shot).

The hyperparameters used for initializing and training comparison models are as follows:

- The EasyOCR implemented **CRNN** (Shi et al., 2016) comparison is trained from a random initialization (as is the default in EasyOCR) for 100,000 iterations on the horizontal English text sequence and horizontal Japanese text sequence synthetic datasets, respectively. The learning rate is fixed at 1.0 with an Adadelta optimizer and the batch size is 128, per the EasyOCR configuration defaults. The architecture uses VGG for feature extraction, a BiLSTM for seq2seq/language modeling, and a CTC loss, as also is the EasyOCR default. A new prediction head is used to match the character set associated with EffOCR for Japanese. The resulting model is then fine-tuned for 30,000 iterations with a batch size of 64, and all other hyperparameters the same, on the benchmark datasets of varying splits.

- The **SVTR** (Du et al., 2022) comparison is first trained from a random initialization for 500 epochs with an Adam optimizer with cosine-scheduled learning rate of 0.001 and batch size of 32 on horizontal English character sequence and horizontal Japanese character sequence synthetic datasets, respectively. All these hyperparameters are PaddleOCR defaults, which are also used for fine-tuning on the benchmark dataset splits.

- The **TrOCR** (Li et al., 2021a) comparison models are initialized from the appropriate vision transformer and language transformer pre-trained encoder and decoder checkpoints: for TrOCR (Base) this is the officially released BEiT (Base) checkpoint and the officially released RoBERTa (Large) checkpoint used by the TrOCR authors for model initialization; for TrOCR (Small) these are similarly the officially released checkpoints for DeiT (Small) and MiniLM used by the TrOCR authors for their model initialization. These checkpoints are exported directly from the TrOCR GitHub repository (Li et al., 2021b) using a modified script originally authored by Hugging Face (Wolf et al., 2020), such that training is possible in native PyTorch with Huggingface model implementations. TrOCR (Base) is trained on the horizontal English synthetic text sequence dataset for 60 epochs at a fixed learning rate of $5e - 7$ with a batch size of 16; TrOCR (Small) is trained for 40 epochs, with all other hyperparameters the same. (The learning rate was selected based on experiments with the validation set.) The resulting models are then fine-tuned with the same hyperparameters on the various benchmark dataset splits.

To evaluate how existing solutions perform when fine-tuned on the EffOCR benchmark datasets, existing pre-trained checkpoints from the EasyOCR CRNN, PaddleOCR SVTR, and TrOCR (Base) and TrOCR (Small) models are fine-tuned on the baseline 70%-15%-15% split of the benchmark datasets. Specifically, the 15% validation set is used for hyperparameter tuning and the 15% test set is used to construct the results reported in the study.

For all comparison models, training hyperparamters are the same as used during the sample efficiency assessments with standardized synthetic pre-training, save that prediction heads for relevant models are left as they are by default. Model initialization differs, accordingly: TrOCR (Base) and TrOCR (Small) use `microsoft/trocr-base-stage1` and
`microsoft/trocr-small-stage1` checkpoints, respectively; EasyOCR CRNN uses the most recently released `japanese_g2.pth` and `english_g2.pth` checkpoints; and PaddleOCR SVTR uses the most recently released `japan_PP-OCRv3_rec_train` and
`en_PP-OCRv3_rec_train` best accuracy checkpoints.

### INFERENCE SPEED COMPARISONS

For digitizing large-scale collections, fast inference on a CPU is necessary, due to the high costs of GPU compute. All comparisons are made on four 2200 MHz CPU cores, selected to represent a plausible and relatively affordable research compute setup. To standardize measurements of speed, each model generated predictions on the same 15% test set. All EffOCR models are implemented with ONNX Runtime for cross-compatibility and speed.

Inference speed is inherently dependent on implementation and it is plausible that the other open-source architectures may be updated in the future to achieve faster inference speeds. A strong correlation between model size and inference speed is apparent and intuitive, highlighting the utility of the EffOCR-C (Small) model for digitizing knowledge - like the Chronicling America collection - at scale.

A random sample of 10 LoCCA scans shows an average of 1944 column x lines per scan (historical newspapers used small fonts and contained few images), which implies the cost at current prices to digitize the LoCCA collection at the line level using GCV would be over 23 million US dollars.

Using FS4 VM instances in Microsoft Azure to process all content in the LoCCA collection for one randomly selected day per decade, on average it took 17.21 seconds to process 1,000 lines with EffOCR-C (small). At current prices, this translates to a cost of \$0.000908 per one thousand lines, as compared to GCV's current prices of \$1.50 (first 5 million units) and \$0.60 (above 5 million units) per thousand lines to process Chronicling America at the line level.

### BENCHMARK DATASET CREATION

Figure S-1 illustrates the documents used to create this study's benchmarks. The OCR systems evaluated in this study take lines (cells in tables or individual lines from columns in prose) as inputs. These segments were created using a Mask R-CNN (He et al., 2017) model custom-trained with Layout Parser (Shen et al., 2021), an open-source package that provides a unified, deep learning powered toolkit for recognizing document layouts. Mask R-CNN was applied to the three Japanese publications considered and to ten different newspapers randomly selected from Chronicling America. Segments were selected at random for inclusion in this study's benchmark datasets. Table S-1 provides dataset statistics.

To create the character region and text annotations, three highly skilled annotators - undergraduate and graduate students - annotated each segment. All discrepancies were then hand checked and resolved by the study authors. Each of the datasets has a 70%-15%-15% train-validate-test split used for baseline evaluations. The validation set was used for model development, whereas the test set was used only once, to create the results reported in this study.

## SUPPLEMENTARY RESULTS

### ABLATIONS

To elucidate which components of EffOCR are essential for its performance, several ablations are examined in Table S-2: using a simple feedforward neural network classifier head for recognition instead of performing k-nearest neighbors classification[2], training with and without hard negatives, disabling training on synthetic data for the recognizer and localizer, and the use of alternative vision encoders. All ablations use a fixed set of hyperparameters that are associated with a specific localizer-recognizer configuration; these hyperparameters are outlined in the sections on Character Localization and Character Recognition.

---

[2]Implicitly, retrieving the nearest neighbor character from an index of offline exemplar character embeddings, as the EffOCR recognizer does by default, is k-NN classification with $k = 1$.

Modeling character-level classification as an image retrieval problem weakly dominates the classification performance when using a standard multilayer perceptron with softmax procedure for classification. OCR as retrieval is chosen as the baseline not only due to its performance, but because it also allows for adding new characters at inference time (just embed a new exemplar character and add it to the offline index) - common in historical and archaeological settings - and because efficient similarity search technologies like FAISS (Johnson et al., 2019) provide fast inference.

Removing hard negatives increases the character error rate substantially, particularly for Japanese, which has many characters with highly similar visual appearances, e.g., some multi-stroke kanji are nearly identical to one another and differ only in the slants of some strokes. Using hard negatives in constrastive training effectively incentivizes the model to distinguish between these very visually similar characters.

Training on only labels from the target documents leads to a large deterioration in performance for Japanese. This is as expected, given that only a fraction of *kanji* characters appear in the small training datasets. The deterioration in performance is modest for English, where there are far fewer characters. The opposite is true for character localization. Localization for English is a harder problem than for Japanese because character silhouettes and aspect ratios are more variable.

Two additional vision transformer encoders are explored: Swin (Tiny) (Liu et al., 2021) for both the localizer and recognizer and ViTDet (Base) (Li et al., 2022) for the localizer and a vanilla vision transformer, ViT (Base), for the recognizer. The performance is similar to the base EffOCR-C and EffOCR-T models.

### OUT-OF-DISTRIBUTION PERFORMANCE

To evaluate how EffOCR does out-of-distribution, we compare EffOCR-Word (Small) to other solutions on a highly diverse, out-of-distribution dataset sampled randomly from 64 randomly selected (out of 638) document record groups in the U.S. National Archives. This is a challenging dataset, with examples shown in Figure S-2.

EffOCR performs similarly to other open-source OCR engines (CER = 12.6), having only been exposed to highly out-of-distribution newspapers during training (Table S-3. GCV performs significantly better than any open-source solution, but with 14 billion documents in the National Archives, the cost of digitizing any appreciable share of them would be astronomical.

### USING EFFOCR TO LIBERATE DATA AT SCALE

EffOCR can convert the publications examined in this study (Jinji Koshinjo, 1954; Teikoku Koshinjo, 1957; Jinji Koshinjo, 1939) into a knowledge graph showing relationships through shareholding patterns, family ownership, financing, boards, occupational histories, family connections, spatial locations, and supply chains.

Figure S-3 provides an illustrative example of one component of this graph, showing supply chain networks in 1956 that were constructed by using EffOCR to digitize the customers and suppliers of Japan's 7,000 largest firms. Fine-grained control through EffOCR allowed detecting an atypical character separating firms in the customer-supplier lists - required for accurate digitization - that other OCR solutions did not systematically recognize, as well as accurate digitization of firm names. Each node in the graph is a firm, whose size is proportional to its degree centrality in the supply chain network. Shading denotes the big-three firms in pre-war Japan (Mitsui, Mitsubishi, and Sumitomo), as well as other firms - comprising Japan's largest conglomerates - targeted by the Holding Company Liquidation Commission in the late 1940s. The graph underscores that the largest pre-war firms remained the most central in Japanese supply chain networks in the 1950s, despite various policies in the late 1940s designed to curb their influence (of Staff, 1945; Commission), 1973; Hadley, 2015; Cohen, 1987).

## SUPPLEMENTARY TABLES

|  | Horiz. Jap. Tables | Vert. Japanese Tables | Vert. Jap. Prose | Chronicling America |
|---|---|---|---|---|
| Train Lines | 1309 | 898 | 459 | 291 |
| Val Lines | 280 | 192 | 98 | 62 |
| Test Lines | 281 | 193 | 100 | 64 |
| Total | 1870 | 1283 | 657 | 417 |
| Train Chars | 3089 | 3296 | 5832 | 7438 |
| Val Chars | 673 | 677 | 1063 | 1708 |
| Test Chars | 682 | 701 | 1111 | 1727 |
| Total | 4444 | 4674 | 8006 | 10873 |

Table S-1: This table reports the number of annotated lines and characters in the training, validation, and test sets of this study's four benchmarks.

|  | EffOCR-C (Base) | Feed Forward Neural Net | Hard Neg. Off | No Synthetic Data Recognizer | No Synthetic Data Localizer | Encoder Swin (Tiny) | Encoder ViT (Base) |
|---|---|---|---|---|---|---|---|
| Horizontal Japanese | **0.006** | 0.006 | 0.041 | 0.594 | 0.009 | 0.009 | 0.010 |
| Vertical Japanese (tables) | **0.007** | 0.010 | 0.087 | 0.700 | 0.016 | 0.016 | 0.010 |
| Vertical Japanese (prose) | 0.030 | 0.038 | 0.076 | 0.788 | 0.032 | 0.036 | **0.027** |
| Chronicling America | **0.023** | 0.037 | 0.045 | 0.027 | 0.068 | 0.025 | 0.037 |

Table S-2: This table provides the character error rate. *Feed Forward Neural Net* models the recognizer as a classification problem with a feed forward neural network, *Hard Neg. Off* does not include hard negatives in recognizer training, *No Synthetic Data* turns off synthetic data training in the recognizer and localizer, respectively, and *Swin (Tiny)* and *ViT (Base)* are alternative vision encoders.

| OCR Model | Character Error Rate |
|---|---|
| EffOCR-Word (Small) | 0.126 |
| Tesseract OCR (Best) | 0.118 |
| EasyOCR CRNN | 0.129 |
| PaddleOCR SVTR | 0.160 |
| Google Cloud Vision OCR | 0.018 |
| TrOCR (Base) | 0.103 |
| TrOCR (Small) | 0.537 |

Table S-3: **Zero Shot Performance on National Archives Dataset.** This table reports off-the-shelf performance of different OCR architectures on a diverse dataset of US National Archives documents.

SUPPLEMENTARY FIGURES

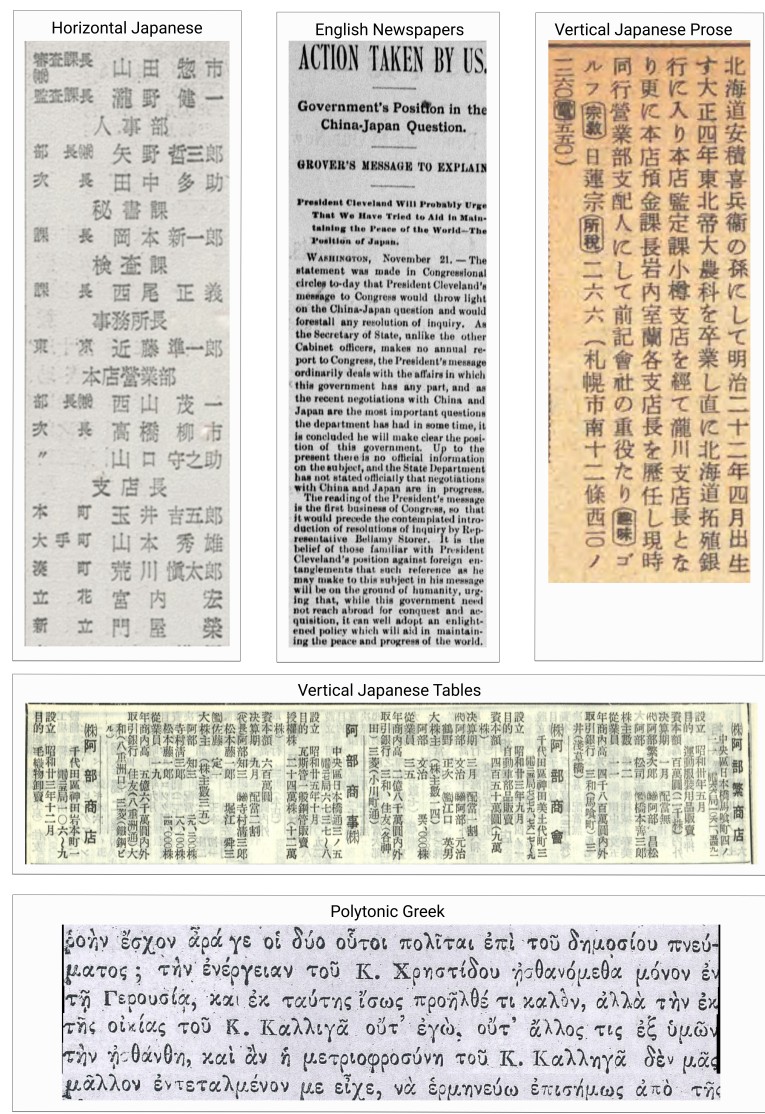

Figure S-1: **Dataset Description.** Representative samples of the publications examined in this study.

| | |
|---|---|
| I am coming to the xonference of County Administrators in Columbia. | I am coming to the xonference of County Administrators in Columbia |
| example of the statements that have emanated from the Bureau of | example of the statements that have emanated from the Bureau of |
| General Laws.   Although the Commissioner of Banks | General Laws. Although the Commissioner of Banks |
| TO WIPE OUT THE SWEAT SHOP, | TO WIPE OUT THE SWEAT SHOP, |
| DUTIES DELEGATED TO SUCH OFFICERS BY LAW. | DUTIES DELEGATED TO SUCH OFFICERS BY LAW. |
| DUTIES OF PATROLS. | DUTIES OF PATROLS. |
| R. W. DODDS, Lt Col, IGD | R, W. DODDS. Lt Col. IGD |
| Chart 1, Page 16 shows "Monthly Revenue Deposited in U. S. | Chart 1, Page 16 shows "Monthly Revenue Deposited in U,S, |
| LOS ANGELES,/2 | LOS ANGELES /2 |

Figure S-2: **Diversity in the National Archives Dataset.** This figure shows examples sampled from the National Archives Zero-Shot evaluation dataset, along with EffOCR predicted transcriptions.

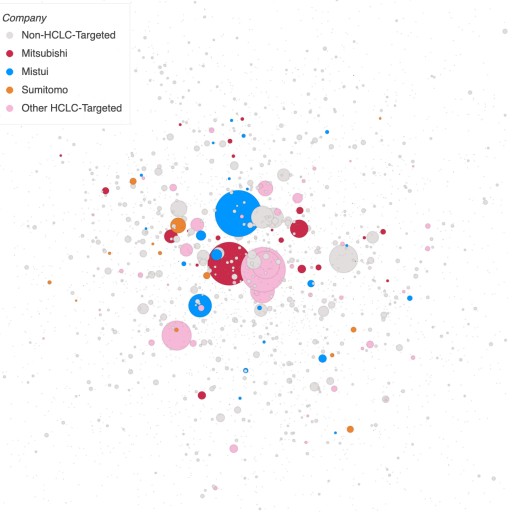

Figure S-3: **Supply Chain Networks (Japan, 1956).** Each node in the graph is a firm, whose size is proportional to its degree centrality in the supply chain network. Shading denotes three of the largest firms in pre-war Japan - Mitsui, Mitsubishi, and Sumitomo - as well as other firms - comprising Japan's largest conglomerates - targeted by the Holding Company Liquidation Commission (HCLC) in the late 1940s.

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
