# OpenReview forum: "Efficient OCR for Building a Diverse Digital History"
_ICLR.cc/2024/Conference — ICLR 2024 Conference Withdrawn Submission_

### Official Review · Reviewer_wGzQ · 2023-10-25

**Soundness:** 2 fair
**Presentation:** 2 fair
**Contribution:** 2 fair
**Rating:** 3
**Confidence:** 4

**Summary:**

This paper proposed a new text recognition method for historical documents. Different from the previous methods that use sequence-to-sequence architectures, it views text recognition as a feature retrieval problem. It does not require a language model or sequential decoding. The experiments on the historical document benchmarks show the effectiveness of the proposed method.

**Strengths:**

1. A new method for recognizing historical documents, which is different from the previous sequence-to-sequence methods.
2. The proposed method shows competing accuracy and efficiency on the historical documents benchmarks, compared with some commercial OCR engines and open-source OCR models.

**Weaknesses:**

I do not think the proposed method is better than sequence-to-sequence methods. The proposed method has some weaknesses:
1. The character detection may cause error, especially for the handwritten texts and blurred texts.
2. It seems that the proposed method need extra annotations (character boxes) to train the model.
3. The retrieval-based method need extra space/memory to store the features.
4. The context information is not utilized, which is very important in difficult cases.

**Questions:**

I did not see the information about the training data. Please remind me if I missed it.

---

### Official Review · Reviewer_rAdB · 2023-10-30

**Soundness:** 3 good
**Presentation:** 3 good
**Contribution:** 3 good
**Rating:** 6
**Confidence:** 5

**Summary:**

The authors propose an OCR model and focus on using encoder-decoder structure to achieve OCRing. Specifically, EffOCR uses historical reading machine combined with OCR to localize individual characters in the document image. It then uses contrastive learning to learn meaningful metric space for character-level OCR. It is worthy to note that the authors are performing OCR at character level and not at word (contiguous characters) level. The authors claim that the model can be easily extensible to other languages. The authors have also discussed the limitations of their proposed model.

**Strengths:**

Some of the strengths of this proposed model are:

1. It can be easily extensible to other languages, especially low-resource or poorly endowed languages.
2. The model does not do actual OCRing like other models. It converts each character to its associated image if there is a match.
3. Because the authors rely on the availability of a character set, the proposed method eliminates the need for annotators.
4. The method is more generic in nature as it doesn’t require a training set.

**Weaknesses:**

There are some weaknesses that the authors can address:

1. In the Introduction section, page 1, “This study shows that”, what study are the authors referring to?
2. Figure 1 could be made more descriptive and legible.
3. In the Results section, page 6, the authors state that “EffOCR learns faster than both Seq2Seq”. This fact cannot be easily drawn from the Table 1. The authors could add an additional line to this to explain what parameters in Table 1 point to this.
4. The authors can test the proposed model on a public database to add a benchmark for the performance since the chosen datasets are not public.
5. How can this be extended to handwritten documents? Does this handle phrases yet?
6. The reliability on character set raises the concern that how would this model handle the absence of characters in a set for a given language?
7. There are some missing references such as in Introduction, page 2, for “Tauschek’s 1920s reading machine”.

**Questions:**

1. Why choose character-level? For searches, most users would prefer phrases, especially when looking into historical documents.
2. Does this model work on handwritten documents? Most historical documents are handwritten.
3. Why was the proposed model compared to Tesseract? Tesseract predicts on sentence-level while the proposed model predicts at character-level.
4. Why is the inference time for smaller models much larger than the bigger models in Table 1?

---

### Official Review · Reviewer_9WAE · 2023-10-31

**Soundness:** 2 fair
**Presentation:** 2 fair
**Contribution:** 2 fair
**Rating:** 3
**Confidence:** 4

**Summary:**

The paper describes an approach for character recognition that is not based on standard seq2seq framework, but on detecting each character in a word, projecting it into a feature space and finding the most similar character from a set of projected training samples. In this wya, the method does not need a large amount of training data and can be used in low resource scenarios/languages. Experimental results are performed on two different languages, Japanese and English, with specific datasets. The proposed method is compared with several commercial and public OCR methods.

**Strengths:**

The proposed method is easily applied to languages and scenarios where a large amount of training data is available. The architecture is very simple and based on standard frameworks. Experimental results show that the method can obtain very good results in a language such as Japanese where symbols correspond to "words", while keeping competitive results in English.

**Weaknesses:**

The method relies on being able to detect and localize characters. In noisy images or in certain types of fonts (including handwriting) this could be difficult and could limit the applicability of the proposed method. A deeper experimental validation would be necessary to show the robustness of the method to noise and detection of individual characters.

In terms of efficciency at test time, the best performing models are big regarding the number of parameters and slower than standard OCR approaches.

The proposed model does not leverage either implicitly or explicitly any language model, as seq2seq models do. Integrating a language model has been shown useful especially in character-based alphabets such as English.

I miss a section on related work contextualizing the proposed method with existing OCR techniques and text retrieval methods.

Some details on the training and evaluation data are not completely clear. For instance, in training the proposed method, have additional synthetic generated samples of each character been used? How many and which types of fonts have been used to generate the samples, if this is the case? Is there an additional evaluation-only datasetset for Japanese as there is for English?

Experimental validation could be extended to using more datasets with larger variability and considering other reent state-of-the-art text recognition methods, including some methods specifically designed for Japanese.

**Questions:**

See above in Weaknesses

---

### Official Review · Reviewer_9D84 · 2023-11-05

**Soundness:** 1 poor
**Presentation:** 1 poor
**Contribution:** 2 fair
**Rating:** 1
**Confidence:** 5

**Summary:**

The paper proposes a character level image retrieval open source OCR architecture EffOCR (efficient OCR), using a contrastively trained vision encoder. The goal is to move away from sequence-to-sequence models requiring huge amounts of training data and compute, to basics of OCR - by localizing characters and recognizing them. Given a generic similarity function can be trained in a contrastive learning architecture, it also scales well to any unknown script or a script with limited annotations. SupCon loss function is used with hard negative mining to further enhance the results. EffOCR toggles between word vs character embedding similarity based on a hyperparameter tuned on validation set.

EffOCR results are shown across two datasets- Japanese historical image datasets and LoCCA. EffOCR provides 0.6% and 0.7% CER for horizontal and vertical Japanese tables respectively 80x better than the next best method evaluated. For English, Google Cloud Vision (GCV) provides better CER. For Greek dataset, however, EffOCR beats GCV by 1.8% in CER.

**Strengths:**

Reducing OCR back to its roots of recognizing characters has been the goal of OCR community since ages. One of the most fundamental approaches in this area is using contrastive learning. The paper has the heart in the right place for choosing the right goal and solution. Localizing characters, however, is very challenging. Latin printed script and Japanese are good examples of the scripts to showcase this as it avoids the character localization problem found in many other scripts (like Arabic, Indic and nearly all handwriting scripts).

Table 1 presents a good overview of the results while Figure 4 illustrates the strength of the approach against very limited training dataset, ousting training-heavy approaches.

It also breaks down the cost of processing a page in GCV vs EffOCR, underscoring EffOCR's low cost to obtain better accuracy.

**Weaknesses:**

The paper has quite a few weaknesses in various areas:
1. Overall presentation: The paper should start with a better lay of the land - state-of-the-art literature, especially in contrastive learning space. It should present the known, recent algorithms that have tried similar approaches. Later, it should go much deeper in the technicalities of the proposed approach - what is the architecture, why is it novel, what advantages does it provide, etc. In dataset section, examples of dataset imagery should be provided with clear presentation of how much text was used for training (in #s) vs testing. In evaluation section, ablation studies is recommended with limitations of the current approach.
It is also suggested to reduce the length of statements to make them more concise and directed.

2. The paper presents one of the many architectures to reduce overall compute from sequence-to-sequence models and presents a well-known fundamental technique in OCR - using contrastive learning. Why is this approach different/novel - should be stated.

3. The datasets used (Japanese historical image datasets and LoCCA) have limited descriptions.

4. The paper compares a specialized model against generic models like GCV, Baidu and Tesseract that have been tuned to work across many scripts and languages. Comparing a specialized model on specific datasets with a pre-trained off-the-shelf network designed for a wide-range of problems is incorrect.

5. Hence, the 80x improvement shown in results is not because the presented approach is novel, but because a) comparative approaches weren't chosen b) specialized models are compared against generic off-the-shelf models.

**Questions:**

1. The paper provides no info on how character localization is performed, which is an essential part of the bigger picture. How do those errors affect the CER? What are typical errors seen in character localization? How do they compare against Japanese and English?

2. Please compare other contrastive learning approaches in OCR with EffOCR, train all of them from scratch and compare them on a larger pool of intended datasets (for all approaches). That would be an apple-to-apple comparison to help evaluate the proposed approach and to understand its limitations and strengths.